Using citizen science in the photo-identification of adult individuals of an amphibian based on two facial skin features

Gould John john.gould@newcastle.edu.au john.gould@uon.edu.au c3129207@uon.edu.au 1
Clulow John 1
Clulow Simon 2
1 Environmental and Life Sciences, University of Newcastle , Callaghan , NSW , Australia
2 Department of Biological Sciences, Macquarie University , Sydney , NSW , Australia
Roberts David
Electronic publication date: 2021 Apr 9
Publication date: 2021
Volume: 9
Electronic Location ID: e11190
Received 2020 Oct 20; Accepted 2021 Mar 9
Copyright: ©2021 Gould et al.
Copyright year: 2021
Copyright holder: Gould et al.
License: This is an open access article distributed under the terms of the Creative Commons Attribution License, which permits unrestricted use, distribution, reproduction and adaptation in any medium and for any purpose provided that it is properly attributed. For attribution, the original author(s), title, publication source (PeerJ) and either DOI or URL of the article must be cited.
License URL: https://creativecommons.org/licenses/by/4.0/

Keywords: Amphibian, Anuran, Biometrics, Mark-recapture, Phenotypic appearance, Skin tubercles, Citizen science, Herpetology

Funding: The authors received no funding for this work.

==============================
Among amphibians, adults have traditionally been identified in capture-mark-recapture studies using invasive marking techniques with associated ethical, cost and logistical considerations. However, species in this group may be strong candidates for photo-identification based on natural skin features that removes many of these concerns, with this technique opening up opportunities for citizen scientists to be involved in animal monitoring programs. We investigated the feasibility of using citizen science to distinguish between individuals of an Australian anuran (the sandpaper frog, Lechriodus fletcheri) based on a visual analysis of their natural skin features. We collected photographs of marked individuals in the field over three breeding seasons using a smartphone device. This photo-database was used to create an online survey to determine how easily members of the general public could photo-match individuals by a comparison of two facial skin features; black banding that runs horizontally above the tympanum and a background array of tubercles present in this region. Survey participants were provided with 30 closed, multiple choice questions in which they were asked to match separate images of a query frog from small image pools of potential candidate matches. Participants were consistently able to match individuals with a low matching error rate (mean ± SD of 26 ± 5) despite the relatively low quality of photographs taken from a smartphone device in the field, with most query frogs being matched by a majority of participants (mean ± SD of 86.02 ± 9.52%). These features were found to be unique and stable among adult males and females. Thus, photo-identification is likely to be a valid, non-invasive method for capture-mark-recapture for L. fletcheri, and likely many anurans that display similar facial skin features. This may become an important alternative to artificial marking techniques, with the challenges of manual photo-matching reduced by spreading workloads among members of the public that can be recruited online.

Introduction

Animal biometrics is an emerging field that involves the identification of species or individuals based on their external phenotypic characteristics, including natural markings or color patterns (Kühl & Burghardt, 2013). It has been used as an effective technique for data collection in ecological procedures such as capture-mark-recapture (CMR), as photographic images of an individual’s unique markings can be cross-matched within a photo-database for detection of recapture events (Williams, Nichols & Conroy, 2002; Pebsworth & Lafleur, 2014). This process has been particularly useful for monitoring species that cannot be easily captured or artificially tagged for identification purposes (Frisch & Hobbs, 2007; Arandjelović & Zisserman, 2011; Hughes & Burghardt, 2015), and has been applied to a diverse number of taxa including mammals (Karanth & Nichols, 1998), large fish (Arandjelović & Zisserman, 2011; Hughes & Burghardt, 2015), crustaceans (Frisch & Hobbs, 2007), and herpetofauna (Gardiner et al., 2014). With the increased affordability and use of smartphone devices that are equipped with cameras, as well as the advent of camera trapping technology, individuals can now be photographed under field conditions and differentiated with very little cost, logistics or expertise required (Wagner et al., 2008; Haddock, Kim & Mukai, 2013; Pebsworth & Lafleur, 2014). Photo-identification is thus becoming an increasingly important ecological tool that is also providing greater opportunities for the use of citizen science in animal monitoring programs (Dickinson, Benjamin & David, 2010).

Nevertheless, some drawbacks have limited the application of photo-based CMR including difficulties in manually processing large image datasets ‘by eye’, particularly those in which markings are only subtly different between individuals, which becomes a time expensive process vulnerable to misidentifications (Katona & Beard, 1990; Bolger et al., 2012; Crunchant et al., 2017). Such obstacles can be overcome using computer-vision techniques that use pattern recognition algorithms, such as ‘hand-crafted’ feature descriptors or deep metric learning, which automatically detect, extract and compare feature information from images uploaded to a photo-database (Van Tienhoven et al., 2007; Takeki et al., 2016; Treilibs et al., 2016; Crunchant et al., 2017). These techniques have been shown to have high accuracy in animal identification and have led to significant labour savings (Morrison et al., 2011). Despite these benefits, most computer assisted systems are only partially automated and still require some degree of manual image processing (Burghardt, 2008). Instead of returning a definitive ‘match’ or ‘no match’ decision, a similarity score is calculated between each image pair, with the strongest matching candidates to the query subsequently needing to be visually inspected in order for a ‘true’ match to be confirmed. It is thus crucial that the feasibility of manual image matching is validated for species prior to the application of photo-based CMR, even for processes that are to become computer-assisted. This challenge of manual image matching, particularly for large databases, may be overcome through the use of citizen science (e.g., Willi et al., 2019), as the speed of matching can be increased by spreading the workload across a large group of people that can effectively be recruited online from any location.

Among amphibians, traditional techniques for identifying individuals include the placement of an artificial visual marker, the removal of toe pads or the insertion of dyes and microchip transponders (Turner, 1960; Brown, 1997; Simoncelli et al., 2005; Bainbridge et al., 2015). These are invasive processes that may influence animal survival, require expertise and are relatively expensive, thereby limiting their widespread use among citizen scientists (Reisser et al., 2008; Sacchi et al., 2010). In contrast, photo-identification is being increasingly used to differentiate amphibians at both the species and individual level (Bradfield, 2004; Church et al., 2007; Gamble, Ravela & Mcgarigal, 2008; Bendik et al., 2013; Sannolo et al., 2016; Konovalov, Jahangard & Schwarzkopf, 2018), and has been shown to have the capacity to outperform traditional marking techniques (Bendik et al., 2013). Nevertheless, the ability for skin features to be used for identification purposes requires investigation to determine if it is a viable alternative that can be performed by volunteers with little prior expertise.

In this study, we examined the feasibility of using citizen science in the visual identification of adult individuals of our model species, the sandpaper frog (Lechriodus fletcheri) based on two facial skin features (banding patterns and tubercles). We asked anonymous participants from the general public, recruited through social media, to visually examine small image datasets and correctly match images of individuals taken at different points in time under natural field conditions from a smartphone device. The main objectives of the study were to determine (i) the potential ease of obtaining sufficiently clear images of individuals from smartphones devices and (ii) the scalability of photo-matching as a technique that has the potential to be used in citizen scientist projects to assist in mark-recapture modelling of anuran populations. Additional objectives were to assess (i) the level of skin feature variability among adult L. fletcheri individuals, and (ii) the stability of skin features in adults over time, as well as to (iii) evaluate the efficacy of using our two facial skin features for accurate photo-identification of an anuran species.

Material and Methods

Study species

Lechriodus fletcheri is a medium sized frog (4–5 cm) found in montane temperate forests along the east coast of Australia (Clulow & Swan, 2018). This species has a prolonged breeding season in the austral spring-summer (September–March) with adults congregating at ephemeral pools during periods of heavy rain to reproduce. Both sexes possess unique facial skin features that have the potential to be used for photo-identification (Fig. 1). A region of black banding that runs horizontally above the tympanum from the corner of the eye to the front leg is a strong candidate marker due to its relatively large size and high level of outline variability between individuals. The skin in this region also has many small epithelial projections (tubercles) that form a unique background pattern which might be an effective secondary feature for photo-analysis. An area of facial skin that included both features was selected as our region of interest (ROI) for visual analysis. All other skin features, particularly dorsal, ventral and leg patterns were not found to be distinct and sufficiently clear for photo-identification purposes.

Figure 1 Target facial skin features of adult Lechriodus fletcheri.

Microscope images of natural skin features present within the chosen region of interest of an male (A) and female (B). Scale bar = five mm.

Frog capture and imaging

This study was conducted within a localized area of the Watagan Mountains (33°00′30.6 S, 151°23′15.7 E, datum: GDA2020), NSW, Australia. During the 2015/16, 2016/17 and 2017/18 breeding seasons, pools located within the study site were routinely surveyed during periods of rainfall for the presence of adult L. fletcheri individuals. Due to the small size of adults, nocturnal activity and preference for wet conditions, individuals had to be hand captured in order to obtain a sufficiently clear and unobstructed view of the facial skin features.

The right ROI of each individual was photographed using an iPhone 6 (Apple Corporation, Cupertino, California, United States) set to manual image capture without flash. To ensure consistency between images, the focal plane of the lens and the lateral side of each frog was kept approximately parallel. This was achieved by gently grasping the back legs so that the thighs were clasped together and the frog kept splayed in a relaxed position, with the index finger kept beneath the belly to keep the body horizontal. Each frog was held approximately 10–20 cm from the lens with the light of a head lamp shone from above the body of the frog to illuminate the skin while avoiding overexposure. Each frog was marked with a passive integrated transponder (PIT) tag prior to release as a secondary method of confirming animal identification. Tags were placed behind the front leg via subcutaneous injection. Photographs were labelled with each individual’s corresponding tag number. Animals were released within a few minutes of point of capture; each photograph was stored in a library along with information pertaining to the date of animal capture. Animals recaptured across each season were processed in the same manner, resulting in a photo-database containing multiple images of individuals recaptured on different dates. In addition, the facial skin features of an adult male and female were photographed using a stereo-microscope mounted DAGE-MTI camera with Leica LAS EZ software V4.0.0 (Leica Microsystems, Wetzlar, Germany), to obtain high resolution images of skin features present within the ROI.

Analysis of variability in skin features

The level of feature variability between individuals was assessed by visually comparing banding regions and tubercles present within the ROI between 10 males and 10 females randomly selected from the database. The temporal stability of these features were also assessed by visually comparing the ROI from individuals which were photographed at different points in time, utilizing the 10 individuals with the longest interval between capture events. In all analyses, ROI’s were extracted from each image by cropping out unwanted sections of skin and background.

Online survey testing capacity of participants to photo-identify Lechriodus fletcheri

An online survey was developed using the ROI photo-library to determine how easily L. fletcheri adults could be photo-matched from images taken at different times during the breeding season by members of the public. We used Facebook’s survey application (Code Rubik Inc, Montreal, Canada) to construct and disseminate the survey. This platform is freely available via any device with access to Facebook and was used given the capacity to reach a large audience with very little cost or logistics. Participation was anonymous and no screening of individuals was performed based on previous photo-matching experience, thereby allowing us to obtain results that would be reflective of the image-matching ability of the general public in a citizen science project.

Our survey consisted of 30 closed, multiple choice questions in which participants were required to match two separate images of the same frog from a pool of images of different frogs. For each question, participants were shown a query image of the right ROI of an individual frog. Six additional ROI images were shown below the query image, five of which were of different individuals that were not a match, and one that was a match showing the same frog but photographed on a separate occasion. Participants were asked to select the matching image amongst the set of six by comparing the banding pattern and positioning of tubercles within the ROI. A sample image of an L. fletcheri adult was provided prior to the survey with instructions on how to compare skin features between images.

The 30 query images were randomly selected from the photo-database, on condition they were of frogs that had been photographed on at least two separate occasions in the field so that matching images of each query frog could be placed into the corresponding answer pool. These matching images were randomly selected from the available images of each query frog remaining within the photo-database. The remaining five non-matching images were also randomly selected from the photo-database, with each answer pool composed of a different combination of non-matching images. Each question was displayed to participants separately, with the order of questions kept constant for each trial. All six answer images were presented together to reduce primacy effects, in which options presented earlier are more likely to be selected (Krosnick & Alwin, 1987). The ordering of images within each question changed for each trial. Participants were only allowed to complete the survey once, with incomplete surveys removed from analysis, which was set up as part of the survey design.

We determined the capacity of participants to successfully match different images of the same frog from small image pools based on the number of questions correctly answered, as well as the time required for participants to complete the survey. Variability in rates of correct image matching between query frogs (i.e., rate of successful matching per question) was also examined, to determine whether some frogs were more easily identified than others. We also examined whether the time taken to complete the survey influenced the proportion of query images that were correctly matched per survey event using a generalized linear mixed effect model (GLMM), with a random intercept to account for differences between participants. Statistical analysis was performed using RStudio version 1.3.959 (RStudio Team, 2020).

Ethics approval

This work was conducted under NPWS Scientific license no. SL101991 and approved by the University of Newcastle Human Research Ethics Committee (approval no. H-2019-0091) and the University of Newcastle Animal Care and Ethics Committee (no. A-2011-138). All experimental procedures were performed in accordance with the Australian code for the care and use of animals for scientific purposes.

Results

A total of 790 photographs were taken over the course of the three breeding seasons. Our database included 606 unique individuals, 15% of which were recaptured and photographed on more than one occasion. The number of days separating capture events varied from a few days to more than a year. Facial bandings used for photo identification were found to be highly polymorphic between individuals of both sexes, along with background arrays of tubercles (Fig. 2). Both features were also found among all individuals captured and clearly identifiable irrespective of sex or skin colouration. Both skin features were also found to be stable over time, with no apparent change after more than a year (Fig. 3).

The mean time difference between obtaining query and matching images of individuals in the field that were then used in the survey was mean ± SD = 15.57 ± 17.90 d. A total of 87 anonymous participants completed the survey; the majority were from Australia (75%) and accessed the survey from a mobile device (77%). The mean time required to complete the survey was mean  ± SD = 14.37 ± 25.03 min, with 90% of participants completing the survey within 20 min (less than one min per question). One participant had an irregularly long survey time (51 h) which was not included in these time estimates.

The average number of query images that were correctly matched ranged from 6/30 to 30/30, with a mean ± SD of 26 ± 5 (Fig. 4). The number of query images correctly matched by a participant was not related to the amount of time taken to complete the survey (GLMM, Z83 =  − 0.003, P = 0.996). Some query frogs were matched correctly more often than others (Fig. 5). The lowest rate of successful matching for a frog was 65% of participants while the highest was 100%, with a mean ± SD of 86.02 ± 9.52%.

Figure 2 Intra-specific variability in banding patterns among adult Lechriodus fletcheri individuals.

Banding patterns of 10 adults of both sexes are shown (A–J = males; 1–10 = females). Banding derived from black and white photographs with background information removed.

Figure 3 Photographs of four adult Lechriodus fletcheri individuals over time highlighting the stability of natural skin features.

The time period between images taken of first capture and subsequent recapture are (A) 299, (B) 383, (C) 86, and (D) 123 days.

Figure 4 Frequency distribution of the number of query images (30 questions in total) correctly matched per survey event.

Note the strong left-skewed distribution, showing participants matched most queries images correctly.

Figure 5 Samples of question and answer images from adult Lechriodus fletcheri used in the online photo-matching survey.

Participants were asked to match a query image of an unknown individual (above the line) with another image taken of the individual at a different point from an image pool of six possible matches (below the line). The correct match for the query individual is the first image below the line, although the order was randomized within questions for the surveys. Samples are from individuals correctly matched often (frog A = 99% and frog B = 97%), or less often (frog C = 68%, and frog D = 66%) by participants.

Discussion

We established that using photographs taken in the field to identify and differentiate between adult individuals of an anuran amphibian based solely on their natural skin marking features is feasible, including the ability of this process of photo-matching to be performed by citizen scientist with very little expertise or prior training. We also met essential criteria for the future application of this technique in capture-mark-recapture studies for our model species; namely demonstration of sufficient inter-individual variation in skin marking features so that individuals can be identified with a high degree of accuracy, and temporal stability of those features so that individuals can be re-identified across subsequent recapture events (Pennycuick, 1978; Marshall & Pierce, 2012).

We found a high level of stable inter-individual variation in facial skin features of both male and female L. fletcheri adults. While participants were asked to match frogs based on a combination of both feature types, the most discernible of these was the region of black banding that runs horizontally above the tympanum. Given its irregular shape, this feature is likely to be the most suitable for manual photo-identification for this species and potentially many other anuran amphibians given the widespread occurrence of this skin feature in this group (e.g., Australian species; Anstis, 2013; Clulow & Swan, 2018). This facial feature may also be a better candidate for inter-individual character discrimination than others previously examined in anurans amphibians such as dorsal patterns, which may be too complex for visual comparison or completely missing in a proportion of individuals of some species (Kenyon, Phillott & Alford, 2009). Differences in the number and positioning of tubercles was also apparent between individuals, though likely more difficult to discern than banding patterns. We suggest that the large number of uniform and repeated features that comprise each array of tubercles across the skin surface would result in a robust visual fingerprint for each individual that would be amendable for computer assisted techniques (Lowe, 1999).

Both facial features in L. fletcheri were temporally stable over at least a 12 month period. This suggests that both are likely to be genetically determined, albeit influenced by environmental factors during development, and permanent (Murray, 1981; Arntzen & Wallis, 1999; Hoffman & Blouin, 2000; Wollenberg et al., 2008), which is critical for studies lasting multiple breeding seasons. Given the short lifespan of L. fletcheri adults (Gould, Clulow & Clulow, 2020), it is difficult to determine whether marking features are stable for longer periods than the recapture intervals obtained in the current study. It is also possible that the appearance of skin features may change if individuals are imaged between different life stages (e.g., sub-adult to adult) or if features are scratched or scarred between capture events (Yoshizaki et al., 2009), although there was no evidence of the latter occurring in our study. Features may also change in appearance due to external sources of variation, such as the method of photographing used. For example, as the skin in the ROI of this species is neither rigid nor flat, the manner in which individuals are held during image acquirement may lead to distortion of features, preventing observers from detecting a true match and leading to possible false rejections. However, this problem was readily mitigated by following a consistent imaging protocol between sampling periods in our study.

The capacity for participants from the public to consistently and correctly identify L. fletcheri individuals based on our target skin features was supported by the results of the online survey. Anonymous participants from the general public were able to correctly match capture (query) and recapture (answer) images of individuals when provided with small image pools, with a majority of participants correctly matching over 85% of queries and a majority of survey questions answered correctly more than 80% of the time. Such rates of successful image classification appear to be comparative to those achieved in the identification of individuals of other animal types (Schofield et al., 2008), and species from large camera surveys (Swanson et al., 2016). Given that participants were provided only minimal training prior to the survey, as well as the short time period taken to answer each question (less than one minute on average), these results suggest that L fletcheri adults can be easily and rapidly photo-identified, at least within small image pools, and that such citizen science programs may be useful in monitoring other amphibian populations.

There was, however, variability in matching score rates between query frogs, with some incurring more false matches than others. Although the reasons for this disparity were not analyzed as part of this study, it is likely that some matches are not as obvious as others, especially if the pool of answer images happen to be of frogs with less distinct features that increase the difficulty in discerning a true match. While 60% of participants correctly matched all individuals, there was also a level of variability in ability for true matches to be detected ‘by eye’. Misidentifications, even if they occur at a low rate, may result in inaccurate parameter estimates in mark-capture-recapture models (Stevick et al., 2001; Yoshizaki et al., 2009). The probability of incorrect matches could be reduced by providing additional training, particularly in the matching of the background array of tubercles that were more difficult to visually assess, by ensuring all images acquired are of sufficient quality before they are uploaded to the photo-database, or by having multiple observers score images and using consensus scores. Indeed, the impact of variable matching ability among participants was not found to be an issue as all query images were overwhelmingly matched correctly when taking into consideration the populous vote.

It is probable that the frequency of correctly photo-matching individuals would decrease as the pool of potential matches increases. Manual photo-matching becomes time consuming and may even become impractical with increasing database size (Katona & Beard, 1990; Speed, Meekan & Bradshaw, 2007; Bolger et al., 2012), a fact that has restricted the use of this technique to small populations (Karanth & Nichols, 1998; Langtimm et al., 2004). Under this scenario, automated approaches may be required. However, it is still important to validate the capacity for individuals from a population to be identified manually. This is because partially automated systems may, on some occasions, only be able to reduce the number of potential matching individuals down to a small pool of candidates with similar marking features, which an observer will then have to sort through to select a true match. This is likely to occur in most populations as some individuals would be expected to share some similar features (Ottensmeyer & Whitehead, 2003), even for those species with high inter-individual marking variability. Validating the ability for correct matches to be manually selected from small image pools is thus an important first step prior to the establishment of computer-assisted methods; a process we have shown could be performed by citizen scientists in future monitoring programs.

We showed that facial marking features in L. fletcheri adults are individually unique and temporally stable, making them effective for photo-identification for conducting non-invasive capture-mark-recapture studies. This process reduces handling times and circumvents the need for invasive tagging techniques with associated logistical, cost and ethical challenges (Reisser et al., 2008). It also has the added benefit of allowing members of the public to become involved in citizen science and conservation programs, either as individuals who are able to acquire images with just the use of a mobile device (under the strict guidance of scientists present), and those who can analyse the data online on their own at any given time. As an increasing number of species require monitoring for their effective management and funding and resources are becoming increasingly spread thin across conservation programs (Margules & Pressey, 2000; Butchart et al., 2010), reliable but less expensive methods of population monitoring are needed. Photographing individuals from mobile devices is likely to help. Although its use requires validation for new species, our results suggest that it is likely to be useful in many other anurans that display distinct facial skin features.

Supplemental Information

Supplemental Information 1 Participant answers to online photo-matching survey

The total number of survey questions answered correctly by each partiicpant has been shown (1= answered correctly, 0= answered incoroectly).

Click here for additional data file.

We thank L. Bainbridge for her advice throughout this study.

Additional Information and Declarations

Competing Interests

Author Contributions

Human Ethics

Animal Ethics

Field Study Permissions

Data Availability

The authors declare there are no competing interests.

John Gould conceived and designed the experiments, performed the experiments, analyzed the data, prepared figures and/or tables, authored or reviewed drafts of the paper, and approved the final draft.

John Clulow and Simon Clulow conceived and designed the experiments, authored or reviewed drafts of the paper, and approved the final draft.

The following information was supplied relating to ethical approvals (i.e., approving body and any reference numbers):

The University of Newcastle Human Research Ethics Committee granted ethical approval to carry out the study (approval reference number: H-2019-0091).

The following information was supplied relating to ethical approvals (i.e., approving body and any reference numbers):

The University of Newcastle Animal Care and Ethics Committee provided full approval for this research (approval number: A-2011-138).

The following information was supplied relating to field study approvals (i.e., approving body and any reference numbers):

The NSW National Parks and Wildlife Service (NPWS), Australia approved a license for this study (Scientific license approval number SL101991).

The following information was supplied regarding data availability:

Raw data, including the number of images correctly matched by survey participants that completed the online photo-matching survey, are available as a Supplemental File.

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
