# Peer review of "Using citizen science in the photo-identification of adult individuals of an amphibian based on two facial skin features"

_PeerJ, doi:10.7717/peerj.11190_

## Round 0.1 · original submission · Major Revisions

Thank you for submitting your manuscript to PeerJ on this interesting subject. It has been reviewed by three experts in this area as well as myself. Two of the reviewers find merit in the paper while one has raised a number of issues. I agree that you need to be careful in your use of the term citizen scientist and this needs to be clarified as are they citizen scientists or study participants. In addition, more consideration of the results needs to be given in the discussion/conclusions to ensure the results support what you are saying. I look forward to seeing a revised version of the manuscript.

Reviewer 1 ·

Basic reporting

The quality of writing is very high. The introduction provides a very readable introduction to the topic for both general and specialist readers, and is well-supported by relevant literature. The paper is well-structured, the figures are all very crisp and clear. I enjoyed reading this paper.

Experimental design

The work falls within the scope of the journal and has clear and well-defined questions. The design and analysis is neat, tidy and straightforward. The methods and results are clearly and concisely presented and the work fills a knowledge gap (if a rather narrow and specialist one).

Validity of the findings

The results are compelling and supported by the data presented. I did wonder about some of the very large standard deviations and units of measurement (see below - do these need checking?), but even if they stand they do not change any of the overall message.

Additional comments

My one minor niggle with the paper is how the work is framed within the role of citizen scientists. Conventionally, citizen scientists are used to collect data at a scale that might be intractable for a small team of professional researchers, i.e. there is some engagement and investment with the project outcomes. In this study, the citizen scientists are used as an anonymous group of helpers to match images, i.e. they are just volunteer helpers without any personal investment in the project outcomes apart (presumably) from a warm feeling that they have helped some scientists. Indeed, I did wonder about what the motivation for the participants might have been here. As this appears to be a rather unusual role for citizen scientists, perhaps this needs unpacking a little more within the narrative and within the framework of the growing citizen science literature.
Minor queries:
Line 223: should this be 15.57 +/- 17.9 seconds? Variation is bigger than the mean – is this correct?
Line 226: Likewise, SD is nearly double the mean – is this correct?
Line 227: 51 min rather than 51 h?
Lines 269-273: Perhaps mention that some programmes allow such distortion to be corrected prior to pattern-matching.
Lines 274-278: Relating to my comment about the citizen science aspect here, are there any advantages to recruiting members of the public over, say, asking a group of students to do the matching? Is the purpose here just to utilize some willing people to do what would otherwise be a routine and mundane task?
Lines 302-307: These are all good points, but the reality here is that if a computer programme was used to narrow down the suite of likely matches, the final identification would be done by an experienced researcher rather than a member of the public. So again, strengthening the rationale for using members of the public might be helpful.
Line 314-315: Indeed, widening the participation of the study so that members of the public can both take the photos and then identify the frogs could increase their investment in the work. Perhaps this needs more emphasis given that they were only involved with the identification phase here.

Reviewer 2 ·

Basic reporting

The manuscript was clear, well written and well presented. The introduction flows well and sets the scene for the research. Figures provided were very helpful in visualising the process.

Experimental design

Line 123 – A sentence on whether these two facial features are the only possibilities, or if there were other features that had been considered and discounted for id purposes, would be really useful to anyone wishing to replicate the research, especially in other anurans. You mention stability of these features in line 220, but this is for your dataset rather than the species. The stability is based around a year, but your images are over three seasons – does this mean that they may change over longer periods, or do the species not live that long? You touch on this in the discussion, but clarification of time scales might help those unfamiliar with the species.
An aside: I ask as I know someone that used photo ID on reptile head markings without considering a paper that showed that scale shapes were stable with age, but not necessarily colour. Is the choice of two to mitigate one or the other changing?

Line 173 – this is a large unsubstantiated assumption. I don’t think that it hinders the research, but it needs clarification. My work with species ID was influenced by face recognition work, and the following may help:
“Passport Officers’ Errors in Face Matching” (White et al., 2014) doi.org/10.1371/journal.pone.0103510, & ‘Super-recognisers’ https://www.theguardian.com/uk-news/2018/nov/11/super-recognisers-police-the-people-who-never-forget-a-face

Validity of the findings

The findings are timely and insightful. By concentrating on one species with a concise task, the findings reveal aspects of species ID that are transferrable, can be replicated & built upon.
The simplicity is refreshing and important for taking to the wider public.

Additional comments

Line 43 - Full stop missing after females
Line 95 – Very glad that you raised the point of survival. These practices, from mutilation to adhesives, are widely applied yet understudied across taxa.
Line 124 – are these features identifiable in different colour morphs (e.g. melanistic, leucistic)?

Line 319 – Can you verifying that these species are not protected in any way. You’ve previously noted that the individuals need to be hand captured – are you expecting the public to do the same when taking photos?

I enjoyed reading the paper and I think that it considers the practical issues surrounding conservation on the ground. Well done.

Reviewer 3 ·

Basic reporting

The title reads like a methods section and does not frame the question.

The abstract is long but fails to report the results (accuracy in matching the correct photograph (26 + 5 / 30 average ~ 86% + 16%), which is annoying as they do not support the interpretation in the abstract ("Survey participants were consistently able to match individuals from small image pools of potential candidate matches with a low matching error rate,").
I would consider an error rate of !4% = 16 % as too high, especially considering the low number of photos to choose from.
Regardless, a brief summary of the numerical results should be in the abstract so the reader can evaluate the evidence.

The rest of the paper is reasonably well written, however the wording is long considering the small results section, based on a relatively small sample size. Much of the introduction and discussion is irrelevant - need to focus on your data / results and papers that have done similar types of analyses.

Perhaps publishable as a short note

Experimental design

Experimental design is flawed due to low number of photographs provided in the study (6 photos / case). I am unaware of any mark and recapture study with a sample size of 6.
This study should have presented a range of photographs that were realistic.

Results difficult to interpret due to low number of volunteers (~ 87 replicates) resulting in high variance.

This type of study should present a range of photographs that were realistic.

Validity of the findings

Conclusions are not supported by the evidence.

The lowest rate of successful matching for individual frogs was 65% of participants while the highest was 100%, with a mean ± SD of 86.02 ± 9.52%.
Combined with:
The poor accuracy in matching the correct photograph (26 + 5 / 30 average ~ 86% + 16%), clearly demonstrates that the error is too high to be reliable or useful in a mark and recapture project.

Furthermore, despite the fact that volunteer labour if free, volunteers took an extraordinary amount of time to match each photo (14.37 ± 25.03 min). How long will volunteers last without quickly falling into volunteer fatigue. Nobody in the 21st century has 20 minutes spare one a regular and repeatable basis.

These results are also weakened by low number of volunteers (~ 87 replicates) resulting in high variance.

In summary, tis lack of support for the conclusions is greatly exacerbated by the poor experimental design - by the experiment only offering 6 choices. In any mark and recapture study there are likely to be at least one order of magnitude greater than this (60 individuals), if not more.

Additional comments

This paper has a fundamental flaw in experimental design that severely limits the interpretation of the results, and the results / evidence does not support the conclusions:

Poor experimental design due to low number of photographs provided in the study (6 photos / case). I am unaware of any mark and recapture study with a sample size of 6. This study should have presented a range of photographs that were realistic.

and a flaw in the interpretation of the results that do not support the conclusions:

The results provide convincing evidence that volunteers show:
• high variability and low accuracy in identifying individual frogs from photographs (26 + 5 / 30 average ~ 86% + 16%),
• a poor individual frog matching rate (86.02 ± 9.52%)
• volunteers took an inordinate amount time / photo (14.37 ± 25.03 min) - and is likely to lead to volunteer fatigue.

These all indicate that using volunteers for this kind of study results in inaccurate results,
despite using unrealistically low numbers of photo options (unrealistic experimental design).
The authors appear to ignore these results and suggest volunteers would be useful (even in this unrealistic scenario).
If we ignore the poor experimental design, these outcomes strongly suggest that volunteers should NOT be used for this kind of study.

---

## Round 0.2 · accepted · Accept

Thank you for making taking on board and considering the reviewers' suggestions. I think it has significantly strengthened the manuscript. I'm happy to now recommend the manuscript be accepted. However please note a couple of small errors that have been highlighted that can easily be dealt with in the proof stage.

Reviewer 1 ·

Basic reporting

The reporting is clear, concise and unambiguous.

Experimental design

This is fine and described clearly.

Validity of the findings

This is fine.

Additional comments

The authors have explained the rationale underpinning the relevance of their work to citizen science more clearly. The relevant sections have been tightened up in the text, and constructive responses have been made to the comments of the reviewers.

Reviewer 2 ·

Basic reporting

I'm happy with the revisions.

Just a couple of typos I noticed:

Line 41 – error rate (mean ± SD of 26 ± 5) is there a % missing?

Line 56 – should read colour

Experimental design

No comment

Validity of the findings

No comment

Additional comments

Thanks for answering the comments in such detail. I look forward to telling my herpy friends about it!